# Canonical NF-κB Pathway as a Central Regulator of Obesity-Associated Inflammation: A Narrative Review

**DOI:** 10.3390/biomedicines13123050

**Published:** 2025-12-11

**Authors:** Flavia-Medana Petrascu, Sergiu-Ciprian Matei, Gheorghe-Emilian Olteanu, Robert Barna, Catalin Marian

**Affiliations:** 1Department of Doctoral Studies, “Victor Babes” University of Medicine and Pharmacy, 300041 Timisoara, Romania; flavia.petrascu@umft.ro (F.-M.P.); cmarian@umft.ro (C.M.); 2Department of Biochemistry, “Victor Babes” University of Medicine and Pharmacy, 300041 Timisoara, Romania; 3Abdominal Surgery and Phlebology Research Center, “Victor Babes” University of Medicine and Pharmacy, 300041 Timisoara, Romania; 41’st Surgical Department, Pius Brînzeu Emergency County Hospital, 300723 Timisoara, Romania; 5Research Center for Pharmaco-Toxicological Evaluations, Faculty of Pharmacy, “Victor Babes” University of Medicine and Pharmacy, 300041 Timisoara, Romania; olteanu.gheorghe@umft.ro; 6Center for Research and Innovation in Personalized Medicine of Respiratory Diseases, “Victor Babes” University of Medicine and Pharmacy, 300041 Timisoara, Romania; 7Department of Microscopic Morphology-Morphopatology, ANAPATMOL Research Center, “Victor Babes” University of Medicine and Pharmacy, 300041 Timisoara, Romania; 8Center for Complex Networks Science, “Victor Babes” University of Medicine and Pharmacy, 300041 Timisoara, Romania

**Keywords:** obesity, NF-κB, inflammation, biomarkers, miRNA

## Abstract

Obesity is characterized by a chronic inflammation driven primarily by sustained activation of the canonical NF-κB signaling pathway. This persistent inflammatory state originates in hypertrophic adipose tissue and progressively affects multiple organ systems, contributing to systemic metabolic dysfunction. This review explores the molecular architecture and regulatory components of the canonical NF-κB pathway and outline how metabolic stressors, cytokines, chemokines, adhesion molecules, and dysregulated microRNAs influence its activity in obesity. Clinically relevant NF-κB linked biomarkers are characterized, those that reflect the degree of inflammation and may support risk stratification in metabolic disease. Finally, we discuss emerging therapeutic strategies, including small-molecule inhibitors, monoclonal antibodies, and miRNA-based approaches designed to modulate NF-κB activity while preserving its essential physiological roles. Together, these insights emphasize the central involvement of NF-κB in obesity-associated inflammation and highlight potential targets for selective intervention.

## 1. Data Acquisition

To explore the role of the canonical NF-κB signaling pathway as a central regulator of obesity-associated inflammation, a structured and comprehensive literature search was conducted across several major scientific databases, including PubMed, Scopus, Web of Science, Google Scholar, MDPI, Nature Publishing Group, Elsevier, and SpringerLink. The search strategy incorporated combinations of relevant keywords such as “NF-κB”, “obesity”, “canonical NF-κB pathway”, “inflammation”, “adipose tissue”, “macrophage polarization”, “inflammatory biomarkers”, and “miRNA regulation”. Boolean operators and, where applicable, MeSH terms were utilized to enhance the precision and scope of the search. No limits regarding publication year were imposed, allowing the inclusion of both foundational studies that defined key aspects of NF-κB biology and more recent investigations that addressed emerging regulatory mechanisms in obesity-related inflammation.

In addition to database queries, reference lists from key articles and authoritative reviews were manually screened to identify additional relevant studies that may not have been retrieved during the initial search. Eligible sources comprised original research articles, experimental studies, clinical investigations, and review papers that addressed the structure, activation mechanisms, regulatory interactions, or functional consequences of NF-κB signaling in the context of obesity or metabolic dysfunction. Only full-length, peer-reviewed studies written in English were included. Publications that lacked mechanistic relevance, which did not involve NF-κB activity in obesity, or did not provide sufficient methodological detail, were excluded.

The data extraction process was guided by a standardized framework designed to capture essential information regarding the molecular structure of the canonical and non-canonical NF-κB pathway, upstream activators relevant to obesity, inflammatory biomarkers, and therapeutic strategies targeting NF-κB signaling. Each eligible study was reviewed in full to ensure accurate interpretation of its findings and to facilitate meaningful integration of its contributions into the broader narrative of NF-κB-driven inflammation. The heterogeneity in study types, experimental models, and outcome measures required a qualitative synthesis rather than a quantitative meta-analysis.

After applying the inclusion criteria, a total of 198 publications were deemed relevant for this review. The findings were synthesized narratively to present a cohesive and comprehensive overview of the mechanisms through which canonical NF-κB signaling contributes to the development and progression of obesity-associated inflammation. This approach enabled the identification of knowledge gaps, emerging regulatory elements, and potential therapeutic targets within the NF-κB signaling network.

## 2. Introduction

Nuclear factor-kappa B (NF-κB) is a pivotal transcription factor involved in regulating numerous genes associated with inflammation and immune responses [1]. Since its discovery NF-κB has emerged as a ubiquitous regulator of cellular homeostasis and a central mediator in the pathophysiology of chronic diseases, particularly metabolic and inflammatory disorders such as obesity [2].

Obesity, recognized as a global health epidemic, involves not only increased adipose mass but also chronic inflammation driven by persistent activation of inflammatory signaling pathways within the adipose tissue. This pro-inflammatory environment contributes to metabolic dysfunction, including impaired glucose and lipid homeostasis [3]. Among the various molecular mediators linking inflammation to obesity, NF-κB has attracted significant attention, facilitated by its role as a transcriptional hub integrating diverse nutritional and cellular stress signals [4,5].

It is clear that metabolic diseases cannot be fully understood or treated without accounting for the role of the immune system and the signaling pathways that connect inflammatory and metabolic cues, NF-κB being emblematic for this intersection [3,5]. NF-κB dysregulation represents a core mechanism through which obesity becomes a systemic disease, affecting not just the adipose tissue but contributing to pathological changes in multiple metabolically active tissues [5,6].

This review explores the multifaceted role of the canonical NF-κB signaling pathway in the context of obesity-induced inflammation, with particular emphasis on the molecular architecture, key regulatory components, clinically relevant biomarkers and potential therapeutic strategies.

## 3. NF-κB Signaling Pathway

NF-κB was first identified in 1986 by Ranjan Sen and David Baltimore as a nuclear factor in B lymphocytes that binds the enhancer of the immunoglobulin k light-chain gene [2,3,7]. This discovery revealed a rapidly inducible transcription factor, later named NF-κB, which opened the door to understanding a pivotal signaling pathway in immunology. Over subsequent years, studies demonstrated the role of the κB enhancer element in B cells and established NF-κB as a key regulator in multiple cellular pathways [7,8].

Further, a family of NF-κB factors encompassing 5 members was identified: NF-κB1 (p105/p50), NF-κB2 (p100/p52), RelA (p65), RelB, and Rel (c-Rel). These are transcription factors that influence the expression of immune and inflammatory genes in response to a wide variety of stress signals. All members contain a conserved Rel homology domain, which enables them to dimerize (formation of homo- or heterodimers), bind specific κB DNA elements and regulate transcriptional activity [8,9,10].

### 3.1. Function and Mechanism (Canonical and Non-Canonical NF-κB Pathways)

The NF-κB signaling pathway is a central regulator of inflammation and immune responses, operating through two main branches: the canonical and non-canonical pathways, as seen in Figure 1 [3,11].

While the canonical and non-canonical NF-κB pathways are often described as distinct, accumulating evidence demonstrates a significant degree of interdependence and cross-regulation. Notably, the canonical pathway is required for the full activation and responsiveness of the non-canonical NF-κB signaling system. This interdependency manifests at both the transcriptional and post-translational levels [11,12,13].

Activation of the canonical pathway typically occurs in response to two broad categories of signals: pathogen-associated molecular patterns (PAMPs) and damage-associated molecular patterns (DAMPs). PAMPs, such as bacterial lipopolysaccharide (LPS) or viral RNA, initiate immune responses primarily during infection [1,3,14]. While DAMPs are the predominant triggers in obesity, where chronic metabolic stress leads to the accumulation of saturated free fatty acids (sFFAs), reactive oxygen species (ROS) and mitochondrial DNA (mtDNA). These endogenous signals activate pattern recognition receptors (PRRs), particularly Toll-like receptors (TLRs) and drive sustained stimulation of the canonical NF-κB pathway perpetuating sterile inflammation [1,14,15].

Under basal conditions, NF-κB activity is kept in check through sequestration of its dimers in the cytoplasm by IκB proteins (inhibitors of κB). The classical IκB family, including IκBα, IκBβ, and IκBε, bind to NF-κB dimers and mask their nuclear localization signals, preventing their translocation to the nucleus. When cells encounter pro-inflammatory stimuli, such as DAMPs, the upstream kinase transforming growth factor-β-activated kinase 1 (TAK1) is activated downstream of receptor signaling and serves as a key mediator that links receptor engagement to IKK complex phosphorylation. The IKK complex of the canonical pathway consists out of: IKK1/IKKα, IKK2/IKKβ and NF-κB essential modulator (NEMO). The activation of this complex leads to the phosphorylation and subsequent proteasomal degradation of IκB proteins, thereby liberating the NF-κB dimers (the most common one is the heterodimer p65/p50). Freed from their inhibitors, the dimers translocate into the nucleus, where they bind to κB sites within the target gene promoters and initiate transcriptional programs involved in inflammation, mainly by allowing the transcription of pro-inflammatory genes [15,16,17,18].

A fundamental requirement for non-canonical signaling is the availability of RelB and p100, the two core components of the RelB/p52 heterodimer that defines non-canonical NF-κB activity. These proteins are not constitutively expressed at high levels but are instead transcriptionally induced by canonical NF-κB activity, particularly through the action of the RelA/p50 heterodimer. Thus, prior or ongoing canonical NF-κB activation facilitates the accumulation of RelB and p100 transcripts, setting the stage for subsequent non-canonical signaling events [14,19,20].

The non-canonical NF-κB pathway is mechanistically distinct from canonical signaling in its activation kinetics, upstream receptors, and transcriptional outputs. The non-canonical NF-κB pathway is activated in a stimulus-specific and tightly regulated manner, primarily in response to a restricted set of ligands from the TNF receptor superfamily. These include CD40 ligand (CD40L), B-cell activating factor (BAFF), lymphotoxin-β (LTβ), receptor activator of NF-κB ligand (RANKL) and TNF-like inducer of apoptosis (TWEAK). In contrast to the canonical pathway, which is rapidly and broadly activated by pro-inflammatory stimuli, the non-canonical pathway is generally engaged during developmental, homeostatic and chronic immune processes [21,22,23,24].

A defining feature of the non-canonical pathway is its reliance on a distinct signaling module. Instead of the canonical IKK complex (IKKα/IKKβ/NEMO), the non-canonical pathway depends on NF-κB-inducing kinase (NIK) and IKK1/IKKα. Upon receptor activation, stabilization of NIK leads to phosphorylation and activation of IKKα. Activated IKKα then phosphorylates the NF-κB2 precursor protein p100, resulting in its partial proteasomal processing into p52 [15,25].

The resulting RelB/p52 heterodimers translocate to the nucleus, where they regulate the transcription of genes involved in lymphoid organogenesis, immune cell survival, chemokine production, and tissue organization. While classical IκB proteins (like IκBα) are not directly involved in this pathway, p100 itself functions as an IκB-like inhibitor of RelB in resting cells [5,24,26].

In obesity, canonical NF-κB signaling plays a central role in shaping and amplifying the non-canonical NF-κB pathway by controlling both the availability of its core transcription factors and the stability of its upstream kinase NIK. Canonical RelA/p50 activation induces the transcription of p100 (NFKB2) and RelB, which are essential components of the non-canonical pathway. This transcriptional induction “primes” cells by increasing the cellular pool of p100 and RelB, thereby enabling stronger RelB/p52 activation once non-canonical receptors such as LTβR, BAFFR, CD40 or RANK are engaged. Beyond supplying these components, canonical NF-κB also regulates the rate-limiting step of non-canonical signaling: the stability of NIK. Under basal conditions, NIK is continuously degraded by a regulatory complex consisting of TRAF2, TRAF3 and the E3 ligase of cellular Inhibitor of Apoptosis Protein 1 and 2 (cIAP1/2), which tags NIK for proteasomal destruction. Canonical pathway activation can influence this machinery in two opposing ways depending on the duration and intensity of inflammatory signaling [26].

During sustained inflammation, a hallmark of obesity, canonical signaling can progressively deplete TRAF2. This TRAF2 is a critical scaffolding protein that anchors TRAF3 and cIAPs to NIK. When TRAF2 levels drops, the entire TRAF2–TRAF3–cIAP degradation system becomes destabilized. TRAF3 is then degraded, cIAPs no longer ubiquitinate NIK, and NIK begins to accumulate in the cell. This shift allows non-canonical signaling to become activated and sustained over long periods [26,27].

Together, these transcriptional and post-translational mechanisms create a coordinated progression from rapid canonical NF-κB activation, responsible for immediate RelA/p50-driven cytokine and chemokine production, to the slower, long-lasting activation of the non-canonical RelB/p52 pathway. This shift is crucial in obesity, where acute inflammatory triggers (such as saturated fatty acids, proinflammatory cytokines) evolve into chronic inflammation, sustained adipose tissue remodeling, and persistent metabolic dysfunction [27,28].

Beyond the intrinsic regulation of canonical and non-canonical NF-κB signaling, adipose inflammation in obesity is further amplified through interactions with parallel stress-activated pathways. An important inflammatory pathway linked to NF-κB in obesity is the MEK/ERK signaling cascade, which is co-activated by the same metabolic and cytokine stimuli and synergistically enhances NF-κB–driven transcription, thereby reinforcing chronic adipose inflammation and insulin resistance [2,3,28].

### 3.2. Regulation of the NF-κB Signaling Pathway

#### 3.2.1. Hormones

The endocrine system, through various metabolic and inflammatory hormones, plays a critical role in modulating NF-κB signaling in obesity. In particular, adipose tissue derived hormones, commonly known as adipokines. These adipokines are represented by leptin, adiponectin and resistin, which act as key regulators at the interface between metabolism and inflammation [29].

Leptin is classically known as a satiety hormone that acts on the hypothalamus to suppress the appetite, it is also involved in proinflammatory responses, as well as angiogenesis and lipolysis [29,30]. In obesity, circulating leptin levels are markedly elevated.

Leptin promotes the activation of the canonical NF-κB pathway in immune cells and the central nervous system (CNS), partly through stimulation of Toll-like receptor (TLR) signaling. Experimental studies have demonstrated that CNS-specific disruption of myeloid differentiation primary response 88 (MyD88), a critical adaptor for TLR-mediated NF-κB activation, protects mice from leptin resistance and diet-induced obesity. On the other hand, overexpression of IKKβ in the hypothalamus (canonical NF-κB pathway activator), impairs leptin signaling, enhances food intake, and promotes weight gain [31,32,33].

Adiponectin, exerts potent anti-inflammatory and insulin-sensitizing effects and is typically downregulated in obesity. Unlike leptin, adiponectin suppresses NF-κB activity by inhibiting IKKβ phosphorylation, thereby preventing nuclear translocation of the p65/p50 NF-κB complex. This leads to reduced expression of downstream inflammatory cytokines such as IL-6 and TNF-α. Restoring adiponectin signaling in obese individuals has been shown to alleviate insulin resistance and reduce systemic inflammation [30,34,35].

Resistin, originally identified in rodent adipocytes and later found to be mainly secreted by macrophages in humans, acts as a pro-inflammatory adipokine linked to obesity and type 2 diabetes. It stimulates the NF-κB pathway via TLR4 and MyD88-dependent mechanisms, leading to increased transcription of IL-1β, TNF-α, and other inflammatory mediators. Elevated resistin levels in obesity contributes to the perpetuation of inflammation and insulin resistance [30,36,37,38].

#### 3.2.2. Inflammatory Cytokines

Proinflammatory cytokines are both products and regulators of the NF-κB signaling, creating feed-forward amplification loops in obesity. NF-κB activation in adipocytes, infiltrating macrophages and endothelial cells leads to the production of more proinflammatory cytokines, therefore perpetuating the inflammation [39,40].

IL-1β is a key autocrine and paracrine signal in adipose tissue macrophages. IL-1β produced by adipose tissue macrophages binds IL-1R on these same or neighboring macrophages, sustaining NF-κB activation and a pro-inflammatory M1 (macrophage type 1) phenotype. IL-1β NF-κB signaling in macrophages induces further pro-inflammatory cytokine and chemokine production. The production of proinflammatory cytokines starts when IL-1β signals through the IL-1R (Interleukin 1 Receptor), binds to this receptor, and recruits the myeloid differentiation primary response protein 88 (MyD88) to the receptor domain. This in turn leads to IKK complex phosphorilation and IκBα degradation, and as a result to nuclear translocation of NF-κB. Through these mechanisms, IL-1β rapidly activates canonical NF-κB in target cells, inducing transcription of inflammatory genes [41,42].

IL-1β is a key autocrine and paracrine signal in adipose tissue macrophages. IL-1β produced by adipose tissue macrophages binds IL-1R on these same or neighboring macrophages, sustaining NF-κB activation and a pro-inflammatory M1 phenotype. IL-1β NF-κB signaling in macrophages induces further pro-inflammatory cytokine and chemokine production. Beyond its local effects, IL-1β contributes to systemic immune activation by promoting monocyte egress from the bone marrow, a process mediated in part by IL-1β-induced upregulation of chemokines such as CCL2 (MCP-1) in adipose tissue. These chemokines establish a chemotactic gradient that recruits circulating monocytes to the inflammed adipose tissue. Once recruited, monocytes differentiate into macrophages and contribute to in situ macrophage expansion, either through local proliferation or further recruitment [43,44,45,46]. IL-1β promotes as well vascular inflammation through NF-κB activation in endothelial cells. Upon binding to IL-1R1, IL-1β activates the canonical NF-κB pathway, inducing the expression of adhesion molecules such as ICAM-1 and VCAM-1. These molecules enhance leukocyte endothelium interactions, promoting the firm adhesion of circulating monocytes and other leukocytes. This adhesion is a critical step preceding transendothelial migration into surrounding tissues, thereby amplifying vascular inflammation [47,48,49].

TNF-α is a key activator of the canonical NF-κB pathway, primarily signaling through TNFR1 (Tumor Necrosis Factor Receptor 1). Upon ligand binding, it starts the activation of IKKβ, IκBα degradation, and nuclear translocation of NF-κB [50].

Under physiological conditions, NF-κB promotes the expression of pro-inflammatory cytokines as well as anti-apoptotic genes, ensuring an appropriate immune response while maintaining cell survival. In obesity, where TNF-α is chronically elevated, signaling predominantly favors sustained canonical NF-κB pathway activation, driving persistent inflammation and contributing to metabolic dysfunction [51,52].

Adipose tissue was the first site where TNF-α was linked to insulin resistance. Through TNFR1 activation, TNF-α stimulates the NF-κB and JNK pathways, leading to serine phosphorylation and degradation of IRS-1, which impairs insulin signaling and reduces glucose uptake in adipocytes. Chronic TNF-α exposure also promotes lipolysis and increases circulating saturated free fatty acids, further aggravating systemic inflammation and insulin resistance [53,54].

TNF-α acts in an autocrine manner on adipose tissue macrophages, where TNFR1 engagement sustains canonical NF-κB activation and drives the continued production of pro-inflammatory mediators. Together with IL-1β, TNF-α forms a positive feedback loop, amplifying the expression of cytokines such as TNF-α, IL-1β, and IL-6, as well as chemokines like CCR2, which promote the recruitment of additional monocytes into adipose tissue [54,55]. TNF-α induced canonical NF-κB pathway activation in endothelial cells, leads also to increased expression of adhesion molecules such as ICAM-1, VCAM-1, and E-selectin, as well as chemokines like CCL2. This NF-κB-dependent transcriptional program promotes endothelial activation, enhancing leukocyte adhesion and transendothelial migration [56,57].

IL-6, unlike TNF-α and IL-1β, does not activate the canonical NF-κB pathway directly through its receptor. Instead, it contributes to inflammation through interaction between the JAK/STAT3 and NF-κB pathways. NF-κB stimulates IL-6 production, and in turn, IL-6 activates STAT3, which can enhance NF-κB activity when both pathways are active in the same cell. This process, known as the “IL-6 amplifier,” occurs mainly in stromal and parenchymal cells, where NF-κB initiates IL-6 expression, and STAT3 helps prolong and intensify the inflammatory response [58,59,60].

Recent studies have highlighted the interplay between IL-6, STAT3, and NF-κB signaling pathways in adipocytes, particularly in the context of obesity-induced inflammation. While IL-6 does not directly activate NF-κB through its receptor, its signaling via the JAK/STAT3 pathway can enhance NF-κB activity in adipocytes. This crosstalk contributes to a positive feedback loop, sustaining the expression of pro-inflammatory cytokines and chemokines that exacerbate adipose tissue inflammation [53,61,62].

Macrophages both produce and respond to IL-6, making it a key modulator of macrophage function in inflammation. IL-6 signaling promotes macrophage survival and supports polarization toward the pro-inflammatory M1 phenotype. In addition, IL-6 acts on circulating monocytes, influencing their recruitment and differentiation within inflamed tissues. Through these mechanisms, IL-6 contributes to macrophage-driven inflammation by sustaining the production of pro-inflammatory cytokines and reinforcing the local inflammatory environment [63,64,65].

The major adipokines and proinflammatory cytokines involved in obesity and their effects on NF-κB signaling are summarized in Table 1.

#### 3.2.3. MiRNA

MicroRNAs (miRNAs) are small, non-coding RNAs approximately 20–25 nucleotides in length that regulate gene expression post-transcriptionally, primarily by binding to complementary sequences on target mRNAs and promoting their degradation or inhibiting their translation [66,67]. Over the last decade, it has become evident that miRNAs are key modulators of both inflammation and metabolism, and their dysregulation contributes significantly to the pathophysiology of obesity [68,69,70,71].

In the context of obesity, specific miRNAs have been shown to influence NF-κB signaling pathways, either by targeting upstream activators such as IKKs and adaptor proteins or by directly affecting NF-κB nuclear translocation and DNA-binding activity [72,73].

Several miRNAs act as negative regulators of NF-κB signaling and are frequently found to be downregulated in obesity. For instance, miR-146a targets IRAK1 (interleukin-1 receptor–associated kinase 1) and TRAF6 (TNF receptor–associated factor 6), pivotal adaptor molecules upstream of IKK activation, thereby dampening NF-κB-driven transcription [74,75]. Likewise, miR-181b limits NF-κB nuclear translocation by repressing importin-α3, while miR-223 directly targets IKKα, attenuating phosphorylation of IκB and subsequent NF-κB activation [76,77]. Notably, miR-29b suppresses NF-κB transcriptional activity by targeting TRAF5, and reduced levels of miR-29b in obesity may facilitate persistent inflammatory signaling. Anti-inflammatory roles have also been proposed for miR-132, which inhibits the TRAF6–TAK1–TAB1 axis, and for miR-21, which suppresses MyD88-mediated NF-κB activation. Interestingly, although miR-132 and miR-21 are upregulated in certain obesity settings, their precise contribution may reflect compensatory or tissue-specific responses aimed at restraining excessive inflammation [77,78,79,80,81,82].

Conversely, several miRNAs enhance canonical NF-κB signaling and are typically upregulated in obesity, thus contributing to sustained inflammatory responses. MiR-155 promotes NF-κB activation by targeting the negative regulator SHIP1 (SH2-containing inositol-5′-phosphatase 1), while miR-221 potentiates pathway activity through suppression of A20 (TNFAIP3, tumor necrosis factor alpha–induced protein 3), a key deubiquitinase that restrains NF-κB signaling [77,83,84,85,86]. MiR-802 activates the IKK/NF-κB pathway by inhibiting NKRF (NF-κB repressing factor), an endogenous NF-κB repressor, further linking adipose tissue dysfunction to inflammation. Additionally, miR-33a amplifies TLR4/NF-κB signaling by modulating lipid raft composition via regulation of cholesterol transporters ABCA1 (ATP-binding cassette transporter A1) and ABCG1 (ATP-binding cassette transporter G1), coupling lipid metabolic disturbances with inflammatory signaling [69,86]. MiR-34a, whose expression progressively increases with diet-induced obesity, enhances NF-κB activity by suppressing SIRT1 (sirtuin 1), a known inhibitor of NF-κB transcriptional output. Its adipose-selective deletion shifts macrophage polarization toward an anti-inflammatory M2 phenotype and mitigates insulin resistance [84].

Table 2 highlights the miRNAs that both influence and are influenced by NF-κB activity in obesity-driven inflammation.

#### 3.2.4. Macrophage Polarization

In lean, metabolically healthy individuals, white adipose tissue (WAT) serves primarily as an energy reservoir and endocrine organ. It is composed of mature adipocytes interspersed with a stromal vascular fraction (SVF) that includes preadipocytes, fibroblasts, endothelial cells, and a modest population of immune cells [90,91]. In this steady state, the immune milieu of WAT is predominantly anti-inflammatory, characterized by regulatory T cells, eosinophils and alternatively activated M2 macrophages. These M2 macrophages secrete anti-inflammatory cytokines (IL-10), supporting tissue remodeling and insulin sensitivity [92,93].

In obesity, adipocytes undergo hypertrophy and hyperplasia. As adipocytes enlarge, they become relatively hypoxic, leading to increased secretion of pro-inflammatory adipokines and chemokines and an altered extracellular matrix. This environment promotes immune cell recruitment and a profound shift in tissue immune composition [94,95,96,97].

Macrophage infiltration into the adipose tissue is a hallmark of obesity-associated inflammation. Chemokines like MCP-1 attract monocytes from circulation, which differentiate into macrophages upon entering adipose tissue [98,99]. These macrophages aggregate around stressed or dying hypertrophic adipocytes, forming “crown-like structures”, where they phagocyte cellular debris and excess lipids [100]. While initially protective, sustained infiltration drives chronic inflammation [101].

In lean adipose tissue, resident macrophages are predominantly M2-like. In obesity, the inflammatory milieu, including saturated fatty acids engaging TLR4 receptors and cytokines like TNF-α, promotes classical M1 macrophage polarization [102,103,104]. This process is critically regulated by activation of the canonical NF-κB pathway, which is triggered by signals such as TLR engagement and TNF-α. Activated NF-κB translocates to the nucleus to induce transcription of pro-inflammatory genes (IL-6, TNF-α, MCP-1), reinforcing local inflammation and recruiting additional immune cells. Moreover, M1 macrophages themselves secrete TNF-α and IL-1β, maintaining NF-κB activation in both macrophages and adipocytes, creating a self-amplifying inflammatory loop [101,105,106,107].

## 4. Canonical NF-κB Pathway: An Autocatalytic Driver of Inflammation in Obesity

The canonical NF-κB pathway plays a crucial role in obesity-induced inflammation, as it serves as a central integrator of metabolic, immune and stress signals within the adipose tissue, as seen in Figure 2A,B [2]. In obesity, triggers such as saturated fatty acids, hypoxia, ROS and adipokines activate NF-κB through TLR4 and cytokine receptors. Once activated, NF-κB drives the transcription of a broad array of pro-inflammatory genes, including TNF-α, IL-6, IL-1β, and MCP-1 [2,108].

This pathway becomes autocatalytic because these NF-κB-dependent cytokines and chemokines reinforce their own production: TNF-α and IL-1β stimulate the activation of the canonical NF-κB in adipocytes, macrophages and endothelial cells, while MCP-1 recruits more monocytes that differentiate into pro-inflammatory M1 macrophages [108,109]. These macrophages further sustain local cytokine release, maintaining and amplifying NF-κB activity [110].

Moreover, dysregulated miRNAs in obesity, with decreased anti-inflammatory miRNAs like miR-146a and increased NF-κB-promoting miRNAs such as miR-155 and miR-34a, remove intrinsic inhibition of this pathway [111,112,113]. Together, these factors establish a chronic, self-perpetuating inflammatory loop, making the canonical NF-κB pathway not just a mediator but a driver of the obesity progression and its metabolic complications [113].

## 5. Target Therapies

Target therapies for NF-κB signaling in obesity has been a long-standing goal, due to its chronic inflammatory activation that leads to insulin resistance and adipose tissue dysfunction. NF-κB is not only a mediator of inflammation but also a fundamental regulator of immunity, cell survival, and tissue integrity. For this reason, complete or systemic inhibition is neither feasible nor desirable. Instead, contemporary therapeutic strategies focus on selective, tissue-specific, or pathway-modulating approaches. The following sections summarize the major therapeutic classes under investigation, outlining their mechanisms, existing data, limitations, and future potential [114].

### 5.1. Small Molecule Inhibitors

Small molecule inhibitors were among the earliest strategies proposed to modulate NF-κB activity in metabolic disease. Initial efforts focused on targeting IKKβ, the central kinase driving canonical NF-κB activation. Genetic studies established that mice lacking IKKβ specifically in adipocytes or in hypothalamic neurons exhibit improved insulin sensitivity, reduced adipose inflammation, and enhanced central leptin responsiveness. These findings created substantial enthusiasm for pharmacological IKKβ inhibition. However, systemic suppression of IKKβ activity quickly revealed major limitations. Because canonical NF-κB is indispensable for host defense, whole-body IKKβ inhibition led to profound immunosuppression. These adverse effects precluded the development of IKKβ inhibitors as a target therapy [115,116].

More refined strategies center on tissue-specific modulation. In particular, hypothalamic inhibition of IKKβ has shown striking metabolic benefit in preclinical models, restoring leptin and insulin sensitivity and preventing diet-induced obesity without causing systemic immune impairment [117]. However, these studies rely on intracerebral drug administration, which is not clinically practical. Efforts are now directed towards nanoparticle or viral-vector–based systems capable of selectively delivering IKKβ antagonists to the central nervous system or to adipose tissue. While promising in concept, such technologies remain early in development and have not yet progressed to clinical testing [118,119].

Another emerging strategy targets the IKK-related kinases IKKε and TBK1. Unlike IKKβ, these kinases are activated downstream of metabolic stress and cytokine signaling, and appear to mediate a compensatory program that suppresses energy expenditure in obesity [120]. Inhibition of these kinases with the small molecule amlexanox improves insulin sensitivity, reduces inflammation, and induces modest weight loss in obese mice [121]. In a Phase 2 clinical trial, amlexanox produced metabolic benefits in a subset of obese individuals characterized by high inflammatory biomarkers, suggesting a potential role for biomarker-guided therapy. Although these benefits are less potent than those observed with GLP-1 receptor agonists, amlexanox represents one of the few NF-κB–modulating small molecules with human efficacy data [122,123].

A number of upstream kinase inhibitors, including TAK1 inhibitors, JAK inhibitors, and experimental NIK inhibitors, have also been explored. TAK1 antagonists effectively block multiple inflammatory cascades, including NF-κB, but their broad activity leads to unacceptable toxicity [124]. JAK inhibitors modulate cytokine-driven NF-κB activation indirectly; this was linked to contradictory effects in obesity [125].

### 5.2. Biological Therapies

Biological therapies, particularly monoclonal antibodies targeting inflammatory cytokines, have reshaped treatment paradigms for autoimmune disorders [126,127]. Their success in rheumatology led to early optimism that similar agents might attenuate obesity-related inflammation by reducing NF-κB activation. However, clinical trials have yielded disappointing results [128,129].

TNF-α was the first cytokine targeted for metabolic indications, based on strong preclinical evidence linking TNF-α to adipose tissue inflammation and insulin resistance [130,131]. Although anti–TNF-α therapies are highly effective in rheumatoid arthritis [127,129], they have consistently failed to improve glycemic control or body weight in obesity and type 2 diabetes [128,129,132]. This failure likely reflects the distributed and redundant nature of inflammatory signaling in obesity: inhibition of a single cytokine leaves many parallel NF-κB–activating pathways fully intact [129,133]. Additionally, TNF-α plays context-dependent roles in adipocyte differentiation and tissue remodeling, and its complete blockade may disrupt physiological processes necessary for metabolic homeostasis [130,134].

IL-1β represents another major NF-κB–linked cytokine. Clinical studies using the IL-1 receptor antagonist anakinra have shown modest improvements in glycemic control and β-cell function in individuals with type 2 diabetes, demonstrating more encouraging results than TNF-α blockade [135,136,137]. However, the metabolic benefits remain limited, and weight loss does not occur [135,137]. The IL-1β–specific monoclonal antibody canakinumab has been shown to reduce cardiovascular events in high-risk patients but does not meaningfully influence diabetes incidence or metabolic function [13,14]. High cost, injection burden, and risk of infection have further limited enthusiasm for IL-1–directed therapies in metabolic disease [135,138].

Blockade of IL-6 signaling with tocilizumab has also been investigated, but IL-6 displays complex context-dependent functions. Clinical outcomes of IL-6 blockade on metabolism are inconsistent, and no clear therapeutic role has been established [139,140]. More pathway-focused receptor modulators, such as lymphotoxin β receptor (LTβR) agonists or antagonists, have shown intriguing preclinical effects on adipocyte differentiation and insulin sensitivity by selectively modulating the non-canonical NF-κB pathway [141,142]. These biologics may eventually provide a more refined approach, but development remains preliminary.

### 5.3. Natural Compounds and Nutraceuticals

Natural compounds offer a distinct therapeutic avenue, with the dual advantages of long-standing dietary exposure and diverse molecular effects. Numerous plant-derived substances attenuate NF-κB signaling, often with fewer safety concerns than synthetic inhibitors [143,144].

Polyphenols found in extra-virgin olive oil, particularly oleocanthal and oleacein, exhibit potent anti-inflammatory activity by preventing nuclear translocation of RelA and by modulating NF-κB–associated microRNAs such as miR-155 and miR-34a [145,146]. Their consumption correlates with reduced inflammatory burden in both human and animal studies, aligning with the well-documented metabolic benefits of the Mediterranean diet [147,148]. Similarly, flavonoids such as quercetin, luteolin, and apigenin suppress NF-κB activation by attenuating oxidative stress, inhibiting IKK activity, and modulating upstream signaling cascades [149,150,151]. Resveratrol, another widely studied polyphenol, suppresses NF-κB by activating SIRT1-mediated deacetylation of p65 and improves metabolic function in preclinical models, though human results remain modest [152,153,154].

Terpenoids such as celastrol have attracted attention due to their profound anti-inflammatory and metabolic effects in mouse models. Celastrol reduces M1 macrophage polarization, improves insulin sensitivity, and markedly reduces body weight in diet-induced obesity [155,156]. However, its clinical translation has been slowed by a narrow therapeutic index and dose-limiting toxicities [155,157].

Other natural compounds, including curcumin, berberine, and various ginsenosides, exhibit broad anti-inflammatory actions and modest metabolic improvements, making them attractive for long-term preventive applications [158,159,160]. Yet, challenges such as poor bioavailability, lack of standardized formulations, and limited high-quality clinical trials continue to impede their integration into evidence-based guidelines [158,161,162].

### 5.4. MicroRNA-Based Therapeutics

MicroRNAs provide an increasingly compelling strategy for modulating NF-κB signaling with greater specificity than traditional inhibitors. Obesity alters the expression of several NF-κB-associated microRNAs, contributing to persistent inflammatory activation [163,164]. Correcting these abnormalities through anti-miR or miR-mimic approaches may reestablish homeostatic control of the pathway [165,166].

Among the most promising targets is miR-802, which is upregulated in obesity and suppresses TRAF3, leading to inappropriate activation of both canonical and non-canonical NF-κB signaling [167,168]. Inhibition of miR-802 reduces chemokine expression, macrophage recruitment, and insulin resistance in mouse models [167,169]. Similarly, miR-210-3p enhances NF-κB activation in adipose-tissue macrophages by repressing SOCS1; anti-miR-210 therapy normalizes macrophage function and improves systemic glucose tolerance in obese mice [170,171]. miR-155, a well-established inflammatory microRNA, promotes NF-κB activation by inhibiting SHIP1 [172,173]. Anti-miR-155 molecules have advanced into clinical trials for hematologic malignancies, raising the possibility of future evaluation in metabolic disease [174,175].

Despite strong preclinical promise, microRNA therapeutics must overcome major translational barriers. These include achieving stable oligonucleotide formulations, ensuring targeted delivery to adipose tissue or macrophages, minimizing off-target effects, and preventing unintended immune activation [176,177]. Delivery systems such as lipid nanoparticles, antibody-ligand conjugates, or engineered exosomes offer potential solutions, but clinical development remains in its infancy [178,179]. As of now, no microRNA-based therapy has been tested in humans for obesity or type 2 diabetes [180].

### 5.5. Lifestyle and Dietary Interventions

Lifestyle-based therapies remain the most reliable means of reducing NF-κB activation in humans [181,182]. Sustained caloric restriction lowers adipocyte hypertrophy and reduces the release of inflammatory cytokines and danger signals that activate NF-κB [183,184]. Exercise is particularly effective, not only because it decreases adiposity but also because it improves mitochondrial function, enhances oxidative capacity, and increases production of anti-inflammatory myokines such as IL-10 [185,186]. Interestingly, acute IL-6 released from contracting muscle exerts anti-inflammatory effects and may counterbalance the chronic IL-6 elevations characteristic of obesity [187,188].

Bariatric surgery represents the most potent intervention for reducing systemic inflammation. Substantial and sustained weight loss dramatically lowers NF-κB activation in adipose tissue, liver, and skeletal muscle while restoring insulin sensitivity [189,190]. These effects appear within days of surgery—well before significant weight loss occurs—reflecting rapid hormonal and inflammatory reprogramming [191,192]. Although surgery is not a direct NF-κB-targeted therapy, its profound impact demonstrates the central role of inflammation in metabolic disease [189,193].

Dietary interventions such as the Mediterranean diet further support NF-κB modulation through high intake of polyphenols, omega-3 fatty acids, and dietary fiber [194]. Omega-3 fatty acids reduce TLR4-driven NF-κB activation, while fiber supports a gut microbiota profile associated with lower endotoxemia and diminished inflammatory signaling [195,196]. Together, these findings illustrate that lifestyle modification can modulate NF-κB signaling through multiple complementary mechanisms [181,197].

### 5.6. Clinical Outlook

Despite decades of research, no NF-κB targeted therapy has yet achieved clinical approval for obesity. The main obstacles include the essential physiological roles of the NF-κB pathway, the difficulty of achieving tissue specificity, and the heterogeneity of human obesity. Future progress will depend on identifying biomarkers that define NF-κB driven obesity phenotypes, refining tissue-targeted delivery platforms, and conducting longer, more carefully stratified clinical trials. Precision medicine approaches integrating genetic, transcriptomic, and metabolomic data will likely guide the next generation of therapeutics aimed at the immunometabolic axis [2,43,197]. Table 3 showcases the current Therapies with NF-κB involvement with a beneficial effect in obesity.

## 6. Conclusions

In obesity, the initial inflammatory response is triggered by DAMPs arising from metabolic stress within adipose tissue. This localized inflammatory response, mediated through canonical NF-κB activation, progressively evolves into a systemic process. Numerous regulatory elements, such as altered hormonal signaling, increased production of pro-inflammatory cytokines, upregulation of adhesion molecules, shifts in macrophage polarization, and dysregulated miRNA expression, further sustain and amplify this inflammatory state.

By evaluating this pathway through specific biomarkers, clinicians may better characterize the inflammatory status of obese patients and implement strategies aimed at preventing the onset of obesity-associated comorbidities. Given the redundancy and complexity of inflammatory networks, combined approaches may ultimately prove most effective for modulating NF-κB signaling in obesity. Pharmacological agents such as amlexanox may achieve greater efficacy when paired with lifestyle interventions that reduce the inflammatory load. Polyphenol-rich diets combined with targeted therapeutics may provide synergistic, multi-level suppression of NF-κB activity. Similarly, future therapies may combine canonical pathway inhibitors with selective non-canonical pathway modulators or microRNA-based interventions, thereby addressing distinct regulatory nodes simultaneously. Integrating anti-inflammatory agents with metabolic therapies such as GLP-1 receptor agonists or SGLT2 inhibitors may also produce additive benefits by simultaneously correcting energy balance, insulin sensitivity, and inflammatory tone.

## Figures and Tables

**Figure 1 biomedicines-13-03050-f001:**
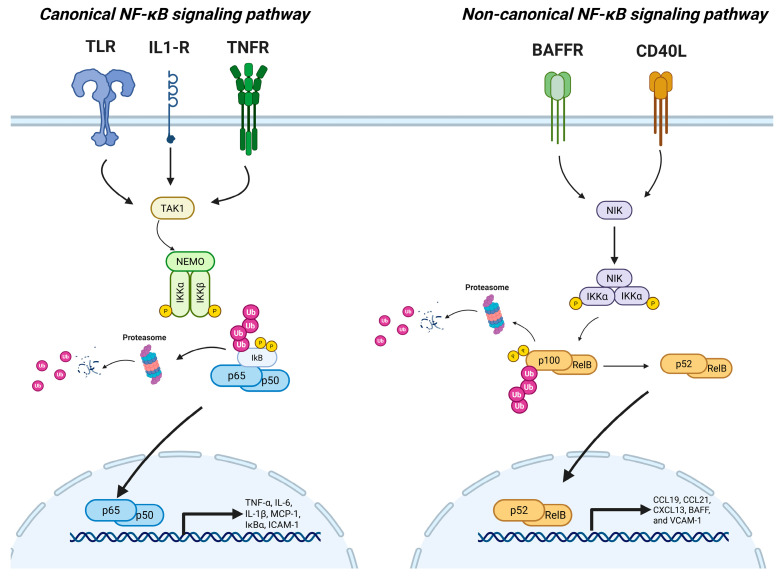
Schematic overview of the canonical (**left**) and non-canonical (**right**) NF-κB signaling pathways. **Left** (Canonical Pathway): Activation of pattern recognition or cytokine receptors such as TLRs, IL-1R, and TNFR initiates the canonical NF-κB signaling cascade. These stimuli activate the upstream kinase TAK1, which in turn activates the IKK complex composed of IKKα, IKKβ, and NEMO (IKKγ). The activated IKK complex phosphorylates IκB inhibitors, targeting them for ubiquitination and proteasomal degradation. This degradation releases the p65/p50 NF-κB dimer, which translocates into the nucleus to induce transcription of pro-inflammatory genes such as TNF-α, IL-6, IL-1β, MCP-1 and ICAM-1. **Right** (Non-Canonical Pathway): Engagement of select TNF receptor superfamily members such as BAFF-R and CD40 activates NIK. Under basal conditions, NIK is rapidly degraded, but receptor stimulation leads to its stabilization. Accumulated NIK phosphorylates and activates IKKα homodimers, which then phosphorylate p100, an inhibitory precursor bound to RelB. This triggers partial proteasomal processing of p100 to p52, releasing the p52/RelB dimer to translocate into the nucleus and activate target gene expression. These genes include those involved in lymphocyte survival, lymphoid organogenesis, and chronic inflammation, such as CCL19, CCL21, CXCL13, BAFF, and VCAM-1. (adaptation after [8] created in BioRender. Anghel, F. (2025) https://BioRender.com/pg0uc1j (accessed on 15 November 2025).

**Figure 2 biomedicines-13-03050-f002:**
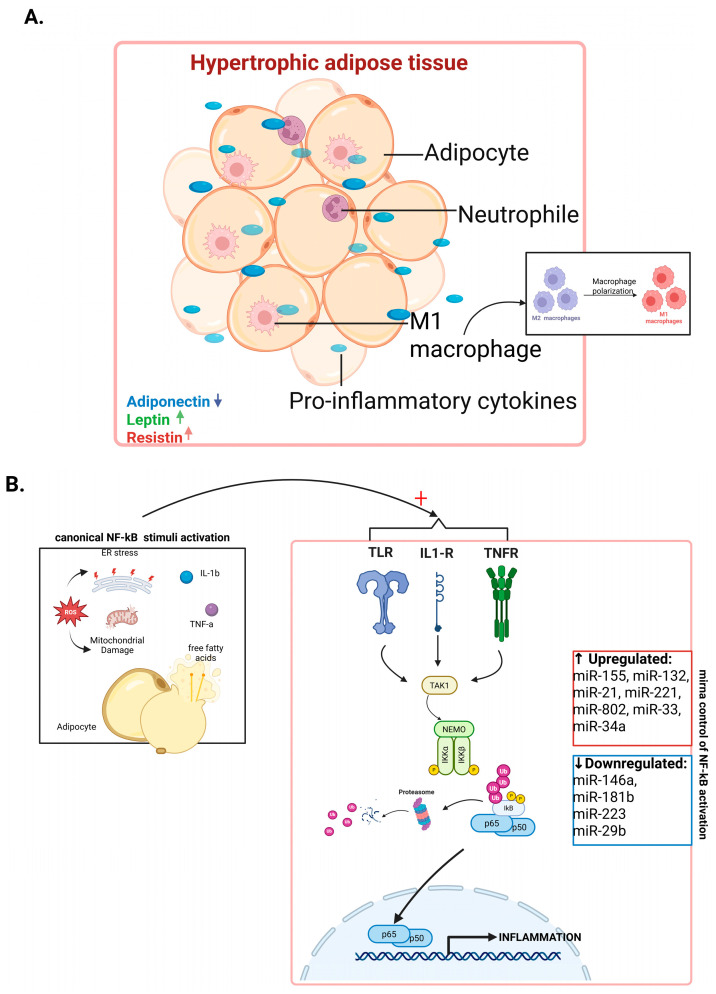
(**A**) Inflammatory remodeling in hypertrophic adipose tissue. In obesity, adipose tissue becomes hypertrophic, attracting more immune cells like neutrophils and pro-inflammatory M1 macrophages. These cells release proinflammatory cytokines, which will worsen local inflammation. At the same time, there is more leptin and resistin, and less adiponectin, which maintains this inflammatory environment (Created in BioRender. Anghel, F. (2025) https://BioRender.com/pg0uc1j (accessed on 15 November 2025)). (**B**) Canonical NF-κB Pathway: An Autocatalytic Driver of Inflammation in Obesity. Stress signals from overloaded adipocytes (ER stress, mitochondrial damage, ROS, free fatty acids) activate the canonical NF-κB pathway via TLRs, IL-1R, and TNFR. This triggers the activation of the IKK complex, IκB degradation and the translocation of NF-κB (p65/p50) into the nucleus, where it stimulates the expression of pro-inflammatory genes. Changes in specific miRNAs further regulate this process: upregulated miRNAs (such as miR-155, miR-21, and miR-34a) amplify NF-κB signaling, while downregulated miRNAs (like miR-146a, miR-181b, and miR-223) remove inhibitory checks of this pathway. Together, these molecular changes reinforce sustained NF-κB activation and chronic inflammation in obesity (Created in BioRender. Anghel, F. (2025) https://BioRender.com/pg0uc1j (accessed on 15 November 2025)).

**Table 1 biomedicines-13-03050-t001:** Adipokines implications in obesity.

Regulator	Type	Primary Source	Receptor/Mechanism	Net Effect on NF-κB	Key Molecular Events	References
Leptin	Hormone (Adipokine)	Adipocytes	TLR → MyD88 → IKK activation	↑ Activation	Elevated in obesity	[29,30,31,32,33]
Adiponectin	Hormone (Adipokine)	Adipocytes	Inhibits IKKβ phosphorylation → blocks p65/p50 nuclear translocation	↓ Inhibition	Downregulated in obesity	[30,34,35]
Resistin	Hormone (Adipokine)	Adipocytes (rodents); Macrophages (humans)	TLR4 → MyD88 → IKK activation	↑ Activation	Elevated in obesity	[30,36,37,38]
IL-1β	Cytokine (Pro-inflammatory)	ATMs; M1 macrophages; adipocytes	IL-1R → MyD88 → IKK → NF-κB activation	↑ Activation	Elevated in obesity	[41,42,43,44,45,46,47,48,49]
TNF-α	Cytokine (Pro-inflammatory)	ATMs; M1 macrophages; adipocytes; endothelium	TNFR1 → IKKβ → NF-κB; parallel JNK activation	↑ Activation	Elevated in obesity	[50,51,52,53,54,55,56,57]
IL-6	Cytokine (Pro-inflammatory)	Adipocytes; macrophages; stromal cells	IL-6R/gp130 → STAT3 → NF-κB amplification	↑ Indirect Enhancement	Elevated in obesity	[53,58,59,60,61,62,63,64,65]

**Table 2 biomedicines-13-03050-t002:** MiRNA regulation of NF-κB in obesity.

miRNA	Effect on Canonical NF-κB Pathway	Expression in Obesity	Reference
miR-146a	Inhibits NF-κB by targeting IRAK1/TRAF6	Downregulated	[43,69,77]
miR-155	Promotes NF-κB activation by targeting SHIP1	Upregulated	[69,77]
miR-132	Inhibits NF-κB via direct targeting of the TRAF6–TAK1–TAB1 axis	Upregulated	[78,79]
miR-181b	Inhibits NF-κB nuclear translocation by targeting importin-α3	Downregulated	[69,80]
miR-223	Targets IKKα, suppressing NF-κB signaling	Downregulated	[70,77]
miR-29b	Targets TRAF5, suppresses NF-κB transcription	Downregulated	[81,82]
miR-21	Inhibits NF-κB by targeting MyD88	Upregulated	[83,84]
miR-221	Enhances NF-κB via suppression of A20	Upregulated	[85,87]
miR-802	Promotes NF-κB activation by inhibiting NKRF, leading to IKK/NF-κB pathway activation	Upregulated	[70,86]
miR-33a	Promotes TLR4/NF-κB activation by regulating ABCA1/ABCG1 (lipid rafts)	Upregulated	[87,88]
miR-34a	Promotes NF-κB activation by suppressing SIRT1	Upregulated	[43,81,89]
miR-126	Inhibits NF-κB activation by targeting VCAM-1 and blocking VCAM-1–dependent NF-κB reinforcement	Downregulated	[43,87]

**Table 3 biomedicines-13-03050-t003:** Most promising NF-κB linked Therapies in obesity.

Therapies	Therapeutic Rationale	Development Stage	References
Lifestyle interventions	Most reliable, safe, and effective NF-κB reduction in humans	Implemented; First-line therapy	[181,182,183,184,185,186,187,188,194,195,196,197]
Bariatric surgery	Most potent anti-inflammatory intervention; rapid and sustained effects	Implemented; Indicated for severe obesity	[189,190,191,192,193]
Amlexanox (IKKε/TBK1 inhibitor)	Only small molecules with human efficacy data	Phase 2 completed; Further optimization needed	[120,121,122,123]
Anti-miR-802	Restores homeostatic NF-κB control; addresses upstream dysregulation	Preclinical; Delivery challenges	[167,168,169]

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
