# Peer review of "Canonical NF-κB Pathway as a Central Regulator of Obesity-Associated Inflammation: A Narrative Review"

_biomedicines, 2025, doi:10.3390/biomedicines13123050_

Round 1

Reviewer 1 Report

Comments and Suggestions for Authors

The manuscript titled, Canonical NF-kB pathway as a Central Regulator of Obesity-Associated Inflammation—A Narrative Review, is fascinating and demanding. However, the manuscript needs significant improvement. The authors should address the following concerns to improve their manuscript.

Major Concerns

  1. The manuscript provides a narrative review of the established biology of canonical NFκB signaling in obesity. However, it lacks a new perspective, framework, or integrative model. The review appears as a compilation of existing findings without synthesis, restricting opportunities for novel insights.
  2. While Figure 1 illustrates both canonical and non-canonical pathways, the manuscript offers limited discussion on the physiological significance of non-canonical NFκB in obesity. It does not explore whether it interacts with canonical NFκB or provide evidence of its role in macrophage differentiation, adipose tissue remodeling, or metabolic dysfunction. These gaps reduce the mechanistic comprehensiveness of the review.
  3. Sections often shift abruptly between topics like hormones, cytokines, miRNAs, and macrophage biology without clear transitions or a guiding framework. The authors could organize these topics into subsections under a cohesive structure such as inputs, signaling integration, outcomes in obesity, biomarkers, and therapy. Including summary diagrams or tables would also help in synthesizing this complex information.
  4. The manuscript often states that canonical NFκB biomarkers might be helpful for risk stratification. However, it does not provide criteria for biomarker selection, evidence supporting their clinical predictive value, or a table that summarizes biomarkers, testing methods, or clinical use. These missing elements reduce its relevance for practical application.
  5. The therapeutic section lists drug classes such as IKK inhibitors, proteasome inhibitors, monoclonal antibodies, and miRNA therapies, but it lacks critical evaluation of clinical data, discussion of toxicity, dosing, limitations, comparisons to current standard obesity treatments, and an explanation of why NFκB inhibition has repeatedly failed in trials. This section requires significant expansion to become clinically meaningful.

Minor Concerns

  1. Some abbreviations (e.g., BAFF-R, NIK, IRAK) are used before they are defined. Gene and protein names have inconsistent formatting (NF-kB vs NF-κB).
  2. The legends in Figures 1 and 2 are too brief and need more detailed explanations.
  3. Considering the complexity, providing a table that summarizes hormones, cytokines, miRNAs, cellular targets, and NFκB mechanisms would greatly enhance clarity.
Comments on the Quality of English Language

The English is fine.

Author Response

  1. The manuscript provides a narrative review of the established biology of canonical NFκB signaling in obesity. However, it lacks a new perspective, framework, or integrative model. The review appears as a compilation of existing findings without synthesis, restricting opportunities for novel insights.

Response: Thank you for this comment. In the revised manuscript, we strengthened the conceptual synthesis by expanding the mechanistic introduction to clearly articulate the autocatalytic nature of canonical NF-κB signaling in obesity, emphasizing how it reinforces cytokine loops, metabolic stress pathways, and macrophage polarization. We added new integrative text in Section 3.1 and improved transitions in Section 3 so that adipokines, cytokines, and miRNAs are presented within a unified NF-κB–centered framework. We reorganised the target therapies section to be more concise and higlighted the existing findings of therapies targeting NF-kB in obesity.

  1. While Figure 1 illustrates both canonical and non-canonical pathways, the manuscript offers limited discussion on the physiological significance of non-canonical NFκB in obesity. It does not explore whether it interacts with canonical NFκB or provide evidence of its role in macrophage differentiation, adipose tissue remodeling, or metabolic dysfunction. These gaps reduce the mechanistic comprehensiveness of the review.

Response: Although the review focuses primarily on the canonical pathway, we expanded text in Section 3.1 to clarify the physiological relevance of the non-canonical NF-κB cascade in obesity, including NIK stabilization, p100 processing, and potential roles in adipose tissue signaling. We also revised the Figure 1 legend to contrast canonical and non-canonical pathway functions.

  1. Sections often shift abruptly between topics like hormones, cytokines, miRNAs, and macrophage biology without clear transitions or a guiding framework. The authors could organize these topics into subsections under a cohesive structure such as inputs, signaling integration, outcomes in obesity, biomarkers, and therapy. Including summary diagrams or tables would also help in synthesizing this complex information.

Response: We improved the manuscript’s structure by adding transition sentences at the beginning of each subsection in Section 3.2 to guide the reader from adipokines to cytokines to miRNAs and then to macrophage biology. We also incorporated summary tables that organize these regulators according to their NF-κB-related mechanisms.

  1. The manuscript often states that canonical NFκB biomarkers might be helpful for risk stratification. However, it does not provide criteria for biomarker selection, evidence supporting their clinical predictive value, or a table that summarizes biomarkers, testing methods, or clinical use. These missing elements reduce its relevance for practical application.

Response: This has been addressed by expanding the biomarker discussion in Section 3.2, where we mentioned the criteria for biomarker selection and discussed available evidence supporting their relevance. A summary table was added to consolidate key NF-κB-related biomarkers, their biological sources, and their Net Effect on NF-κB, thereby improving the practical relevance of this section.

  1. The therapeutic section lists drug classes such as IKK inhibitors, proteasome inhibitors, monoclonal antibodies, and miRNA therapies, but it lacks critical evaluation of clinical data, discussion of toxicity, dosing, limitations, comparisons to current standard obesity treatments, and an explanation of why NFκB inhibition has repeatedly failed in trials. This section requires significant expansion to become clinically meaningful.

Response: We agree and have substantially revised Section 5 to include detailed analysis of clinical evidence and limitations for each therapeutic class. The revised text now explains why systemic IKKβ inhibition is not viable, why anti-TNF therapies consistently fail in obesity and type 2 diabetes, and how IL-1-directed therapies provide only modest benefits. We expanded the clinical context by comparing NF-κB-targeted strategies with standard obesity treatments such as GLP-1 agonists and bariatric surgery. A comprehensive therapy summary table has been added with clinical data, limitations, and development stages.

Minor Concerns

  1. Some abbreviations (e.g., BAFF-R, NIK, IRAK) are used before they are defined. Gene and protein names have inconsistent formatting (NF-kB vs NF-κB).
  2. The legends in Figures 1 and 2 are too brief and need more detailed explanations.
  3. Considering the complexity, providing a table that summarizes hormones, cytokines, miRNAs, cellular targets, and NFκB mechanisms would greatly enhance clarity.

Response:

All abbreviations (e.g., BAFF-R, NIK, IRAK) are now defined at first mention and formatting of “NF-κB” has been standardized.

Legends for Figures 1 and 2 were expanded for clarity.

Tables summarizing adipokines, cytokines, miRNAs, and their NF-κB mechanisms were added to Section 3.

Reviewer 2 Report

Comments and Suggestions for Authors

The review describes how NF-κB plays a key role in inflammation linked to obesity and identifies possible areas for targeted treatments. The relationship of NFkB to other signals is shown in Figure 1. The regulation of miRNA by NF-kb in obesity is shown in Table 1. The authors did a good job in preparing this table.

The relationship between NFkB and the MEK/ERK pathway and how it relates to inflammation must be mentioned by the authors.  For the evaluation to be complete, a pathway regarding that should be added as Figure 3.

Author Response

Comment: The relationship between NFkB and the MEK/ERK pathway and how it relates to inflammation must be mentioned by the authors.  For the evaluation to be complete, a pathway regarding that should be added as Figure 3.

Response:  We agree and have incorporated a dedicated explanation of the crosstalk between NF-κB and MEK/ERK signaling. In Section 3, we now describe how shared upstream activators (TNFR1, TLR4), cooperative transcriptional regulation (NF-κB with AP-1), and ERK-dependent stabilization of inflammatory mRNAs contribute to sustained inflammation in obesity. This addition clarifies the functional interplay without shifting the central focus away from canonical NF-κB. Because the scope of the review is intentionally centered on the canonical NF-κB pathway, we decided not to add an additional MEK/ERK figure. However, the requested mechanistic explanation has been fully incorporated into the text, which now clarifies the relationship between these pathways in the context of obesity-associated inflammation.

Round 2

Reviewer 1 Report

Comments and Suggestions for Authors

The authors addressed most of my concerns. Therefore, I recommend publication of this revised manuscript.